# The Effect of Nitrogen Supply on Water and Nitrogen Use Efficiency by Wheat–Chickpea Intercropping System under Rain-Fed Mediterranean Conditions

Fatma-Zohra Bouras [1], Salah Hadjout [2], Benalia Haddad [1], Asma Malek [1], Sonia Aitmoumene [1], Feriel Gueboub [1], Luiza Metrah [1], Bahia Zemmouri [1], Omar Kherif [1], Nazih-Yacer Rebouh [3] and Mourad Latati [1,3,*]

1   Laboratoire d'Amélioration Intégrative des Productions Végétales (C2711100), Département de Productions Végétales, Avenue Hassane Badi, Ecole Nationale Supérieure Agronomique (ES1603), El Harrach, Algiers 16200, Algeria
2   Centre de Recherche en Aménagement du Territoire—CRAT, Campus Universitaire Zouaghi Slimane, Constantine 25000, Algeria
3   Department of Environmental Management, Peoples' Friendship University of Russia (RUDN University), 6 Miklukho-Maklaya Street, 117198 Moscow, Russia
*   Correspondence: m.latati@yahoo.com; Tel.: +213-671606269

**Abstract:** It is well known that legume–cereal intercropping systems are more efficient in terms of resources use, in particular nitrogen (N) and water. However, the response of this cropping system to water and N co-limitation was poorly studied in most of the recent field researches. The present study aims to assess the relationship between N and water use efficiency (NUE and WUE) by chickpea–durum wheat intercropping under contrasted climate and N supply conditions. Field experiments were carried out during the 2021/2022 growing season, in three sites located at both sub-humid and semi-arid regions. WUE, NUE, aboveground biomass, grain yield and crop physiological parameters for either intercropped chickpea or durum wheat were assessed and compared to the respective measurements in monocultures among all N-fertilizer level × site treatments. The results showed that WUE relative to grain yield ($WUE_{GY}$) and biomass ($WUE_{YB}$) were significantly higher in sole cropped wheat under the conditions of the three studied sites, except for $WUE_{YB}$ in S2, in which intercropping increased $WUE_{YB}$ by +0.46 and +1.03 kg m$^{-3}$, as compared respectively, to monoculture under low application of N fertilizer. As compared to chickpea monoculture, intercropping increased $WUE_{GY}$ by more than 0.30 and 0.57 kg m$^{-3}$ under semi-arid conditions (S1 and S3) over three N-application doses, and by more than 0.18 kg m$^{-3}$ under sub-humid conditions (S2). Simultaneously, NUE was significantly increased by intercropping, where in the mixed crop the highest values were noted as compared to sole-cropped durum wheat and chickpea. However, reducing the N-application dose leads to a gradual increase in NUE by more than 4.44 kg kg$^{-1}$. As a consequence, intercropping enhanced protein accumulation in the grain yield of mixed crops by more than 30 kg ha$^{-1}$ as compared to sole-cropped durum wheat, in particular under moderate N-application and sub-humid climate. Indeed, average chlorophyll content was increased (7.8%) in intercropped durum wheat under all applied N-doses in sub-humid conditions. Rain-fed chickpea–wheat intercropping promotes an improvement in growth and yield quality thanks to simultaneous optimization of water and N use under low and moderate N-application in both semi-arid and sub-humid climates.

**Keywords:** chickpea; intercropping; water use; protein; nitrogen fertilizer; climate

## 1. Introduction

The limits of traditional agricultural production systems (i.e., crop rotation, fallow-crop rotation, extensive monoculture and cover crop), raise the question of improving their efficiency, in particular, by developing agricultural practices such as "low input farming systems" [1]. In order to support better management of cropping systems in the

Mediterranean region, new cropping practices based on agro-ecological intensification through species diversification are proposed at both plot and farm scales [2]. Cereal–legume intercropping is one of the most productive and sustainable cropping practices thanks to its facilitation of resources use, which allow to improve growth, soil fertility and yield [3,4]. In addition, legumes are introduced in mixture with cereal crops to reduce soil erosion, and increase land and resources (i.e., water and nutrients) use efficiency [5,6]. However, crop production and quality of intercropped cereals and legumes are generally affected by various agronomic variables. Thus, intercropping can affect crop yields among its practice and management such as: component crop density, spacing and arrangement of intercropped plants, cropping time relative to each intercropped species and efficiency in use of water and nutrient inputs by the two intercropped species [7]. Because of this, the adoption of these diversified cropping systems as an innovative agricultural practice raises the question of their acceptability in the agronomic sector, in which they could promote the resilience and sustainability of cereal crops that are widely practiced in either intensive or extensive monoculture systems [8,9]. Recent studies have identified the legumes-based intercropping system as one of the most sustainable practices for boosting agro-ecosystem services, in particular under low-input farming systems [10]. For example, previous recent field researches have reported that growth and yield advantage are confirmed in wheat–legume-based intercropping under moderate N and water limitation [6,8].

Given soil nutrient depletion and water scarcity for agriculture in the Mediterranean regions, crop diversification (e.g., mixture crop, cover cropping, agroforestry and rotation) provides sustainable solutions to the problem of the intense use of natural resources in agriculture [11]. In Algeria, the fallow–cereal rotation remains the most frequent practice for cereal production. Replacing fallow practice has become a necessity, to guarantee food security for the Algerian population that is expected to reach 60 million in 2050 [12]. Legumes can play a key role in the resorption of large agricultural areas in fallow by their establishment in rotations and/or in association with cereals [13]. In this context, recent field researches were performed in cereal agrosystems in semi-arid and sub-humid regions in Algeria, to provide better decisions that can help farmers to replace fallow rotation by either legumes or cereal–legume intercropping systems [13–16]. These last studies reported mainly that all assessed intercropping systems (i.e., cowpea–maize, common bean–maize, faba bean–wheat and chickpea–wheat) were more efficient in terms of (i) increasing water and nutrients acquisition (e.g., nitrogen, phosphorus and calcium), (ii) improving growth and yield of intercropped cereals and (iii) enhancing land use efficiency as compared to the respective fallow and monoculture farming systems.

In low input farming systems, one of the most important advantages of intercropping is the ability to enhance both acquisition and use of available resources leading to increased productivity compared to cropping alone. This provides yield advantages due to the fact that the growth resources, such as light, water and nutrients, are more fully acquired and/or converted to biomass by intercropping as compared to monoculture [17]. In recent years, cereals crop diversification by using legumes-based intercropping systems is strongly supported as one of the solutions to enhance WUE and NUE under low-input agriculture. For example, both crop productivity and WUE were significantly increased in intercropped wheat with faba bean thanks to reducing direct soil evaporation and efficient sharing of water for plant transpiration [18]. A recent field study reported a significant increase of WUE in intercropped maize with durum wheat and pea as a result of increase in yield per unit of water supplies [19]. In terms of NUE, intercropped legumes can improve N acquisition by the respective intercropped cereals thanks to symbiotic fixation of atmospheric nitrogen ($N_2$) [20]. Moreover, the legume-based intercropping system was identified as an efficient practice for enhancing agroecosystem performance and resilience by limiting the use of synthetic N-fertilizers as a consequence of enhancing NUE [21]. They can also reduce water pollution by decreasing nitrate leaching ($NO_3$) when they are cultivated as intercrops with cereals [22].

The increase of cereal and legume yield in monocultures through improved water and N management was much explored in the literature [23]. This was probably due to the good understanding of the N and water use by the growing crops among the fundamental processes (i.e., plant growth, photosynthesis, respiration, yield formation) in crop production [24]. Interaction between water and N availability may lead to either negative or positive effects on crop yield, WUE and NUE [23,25]. For example, optimal water availability may contribute to N-losses via $NO_3$ leaching and denitrification, while N inputs applied through N-fertilizer could stimulate growth, but may also lead to early exhaustion of soil water, especially in semi-arid regions [24]. Nevertheless, the interaction effect between WUE and NUE by intercropped legumes and cereals is poorly documented in the literature, especially under contrasted conditions of either climate or N-application. Thus, the growth and yield responses to water and N availability was separately studied in the most of the field research that was focused on cereal–legume intercropping systems [16,26]. Hence, the simultaneous diagnosis of both positive and negative interactions between NUE and WUE may provide a quantitative understanding of water and N use, by intercropped cereals and legumes. This probably led to simultaneous optimization of water and N use by the two intercropped species, over a wide range of climate, soil fertility and also water and N input levels.

This study aims to assess the growth and yield responses to water and N availability in rain-fed chickpea–durum wheat intercropping in both sub-humid and semi-arid climate conditions. The main objective of this research investigation is to assess simultaneously WUE and NUE in intercropped species as compared to their respective monocultures, and understanding their interactions in crop growth and yield under contrasted supply of N-fertilizers. The present study addresses three specific research questions: (i) In which climate conditions do the combined effect of cropping system and N-application level significantly affect growth parameters and yield of chickpea and durum wheat? (ii) Can intercrops enhance N acquisition and water use per unit of land area as compared to either sole cropped chickpea or durum wheat? (iii) In which conditions of climate and N-availability (N-fertilizer input) are both WUE and NUE simultaneously increased in intercropping systems?

## 2. Materials and Methods

### 2.1. Experimental Sites and Climate Description

This study was carried out during the growing season from December 2021 to June 2022, in open fields at three experimental sites (S1, S2 and S3). The S2 site is located in the Metidja region, northeast of Algiers (36°42′ N, 3°09′ E). However, the S1 and S3 sites are located respectively, in the center (36°06′ N, 5°20′ E) and south (35°53′ N, 5°39′ E) of the Setif region. The soil physical and chemical properties are shown in Table 1. The three study sites are located in contrasting conditions of soil characteristics and climates. The meteorological conditions are shown in Figure 1, indicating the cumulative rainfall and average temperature, during cropping time cycle (from December 2021 to June 2022), as well as an inter-annual average of precipitation and temperature for the period ranging from (1990–2020).

Figure 1 shows that the S1 site recorded a cumulative rainfall of 290 mm during the growing season, while the cumulated precipitation in S2 and S3 sites was 387 and 219 mm, respectively. Temperatures ranged from 12.35 to 15.23 °C, with the highest temperature noted in June and the lowest in January (26.12 and 5.12 °C, respectively). Figure 1 also shows the monthly inter-annual mean values of precipitation and temperature observed over the period 1990–2020. Data show the same trend of distribution in both temperature and precipitation, but with overall less precipitation and higher temperatures. This may be related to climate change and the year's drought.

**Table 1.** Soil physico-chemical proprieties of the three studied sites.

| Soil Proprieties | S1 | S2 | S3 | *p* Value |
|---|---|---|---|---|
| Clay (%) | 42.53 [b] | 56.5 [a] | 49.22 [c] | ≤0.001 |
| Loam (%) | 35.78 [a] | 35.2 [a] | 34.83 [a] | 0.9 |
| Sand (%) | 21.67 [a] | 8.4 [b] | 15.95 [a] | 0.01 |
| CaCO3 (%) | 21.94 [a] | 1.1 [b] | 20.58 [a] | ≤0.001 |
| OM (%) | 1.24 [b] | 1.8 [a] | 1.90 [a] | 0.004 |
| Total N (g kg$^{-1}$) | 1.36 [b] | 1.36 [b] | 2.38 [a] | ≤0.001 |
| pH | 8.37 [a] | 7.9 [b] | 8.30 [ab] | 0.02 |
| N-available (mg kg$^{-1}$) | 17.25 [ab] | 11.21 [a] | 37.20 [b] | ≤0.01 |
| P-available (mg kg$^{-1}$) | 8.75 [ab] | 12.51 [a] | 4.85 [b] | 0.02 |
| K (meq 100 g$^{-1}$) | 0.58 [a] | 0.39 [b] | 0.45 [b] | 0.05 |

Data are means ± standard error of 4 replicates. Mean values labeled with the same letter were not significantly different at *p* < 0.05.

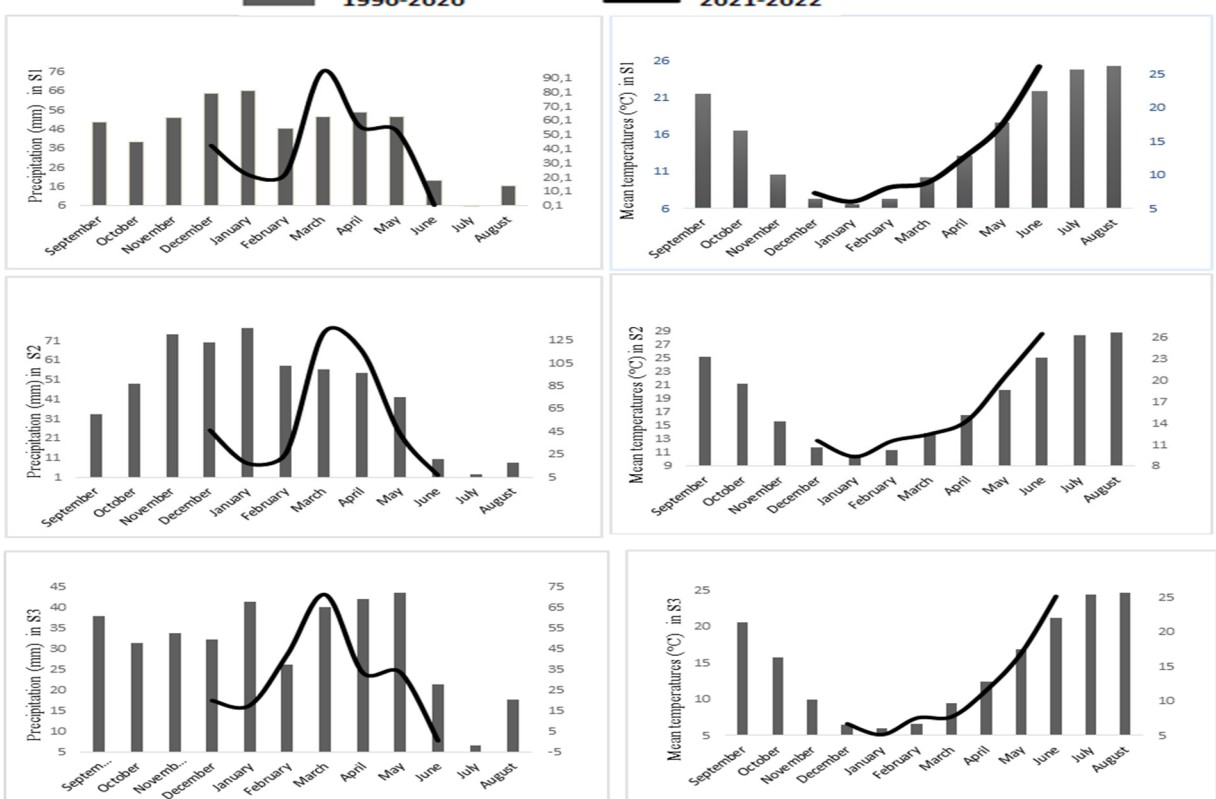

**Figure 1.** Monthly temperatures and precipitation over the 2021–2022 growing season and inter-annual mean values recorded over the period 1990–2020 at the three experimental sites (S1, S2, and S3). Meteorological data were collected from the website of the National Office of Meteorology in Setif (Algeria) (https://www.infoclimat.fr/observations-meteo/archives/1er/janvier/2019/setif/60445.html (accessed on 12 August 2022)).

Moreover, soil physical and chemical proprieties of each studied site are given in Table 1. Soil texture was significantly different among the three sites, except for the loam proportion (Table 1). Table 1 shows also that all studied sites are characterized by clay–loam soil, in which the clay rate varied from 42.53 (S1) to 56.5% (S2). However, the soil of the Setif region (S1 and S3 sites) was richer (15 and 21%, respectively in S1 and S3 sites) in sand than that of the Metidja region (8% in S2 site).

The soil pH varied from 7.90 to 8.37 and was much more alkaline in the experimental sites of the Setif region (S1 and S3 sites). The soil in the S1 and S3 sites was also classified as calcareous soil, with a significantly higher CaCO$_3$ content (from 20.58 to 21.94%), compared

to the S2 site (1.1%). Total nitrogen content also differed significantly between three sites. The highest nitrogen (N) content was observed in the S3 site (2.38 g kg$^{-1}$), while the lowest value (1.36 g kg$^{-1}$) was observed in both S1 and S2 sites. The same trends were found for soil organic matter, with a significant difference between the three sites studied (Table 1). In terms of soil mineral availability, the lower phosphorus (P) availability was noted in the soil of S3 (4.85 mg kg$^{-1}$) where soil N-available was noted to be the greater value (37.20 mg kg$^{-1}$) as compared to that in S1 (17.25 mg kg$^{-1}$) and S2 (11.21 mg kg$^{-1}$) sites. For soil potassium (K) concentration, the highest value was observed in S1, while the lower values were found in both S2 and S3.

## 2.2. Plant Material and Crop Management

The study was carried out with two varieties of both chickpea cultivar (*Cicer arietinum* L.cv. FLIP 90/13 C) and durum wheat cultivar (*Triticum turgidum durum* L.cv. VITRON). These two varieties are commonly cultivated by Algerian farmers in either sub-humid or semi-arid regions. Two combined factors are included in the experimental design: (i) N-fertilizer dose with three N-levels, equivalent to 30, 60, and 100 units ha$^{-1}$ (i.e., N-30, N-60, and N-100) and (ii) cropping system factor that correspond to chickpea monocrop, durum wheat monocrop and chickpea–durum wheat intercropping. The experiment plot was divided into nine sub-plots (treatment) within three replicates for each sub-plot. The treatments were arranged under a randomized complete block (RCB) design where the total covered area about 600 m$^2$ for S1 and S3 sites, and 1311 m$^2$ for S2 site including borders area. The planting density was chosen according to local standard cultural practices that are commonly adopted by farmers. There were a total of 350 plants per m$^2$ for durum wheat monocrop, 30 plants per m$^2$ for chickpea monocrop, while it was 150 and 18 plants, respectively for intercropped durum wheat and chickpea. The inter-row distance was 17 cm for monocropped durum wheat, while it was 30 cm for chickpea in both monoculture and intercropping system. N-application was applied with urea (46-0-0) during two periods of the cropping cycle. The first N-supply was applied at the beginning of durum wheat tillering (30 units/ha$^{-1}$ was applied for each N-level treatment), while the second one was applied at beginning of stem elongation (0, 30 and 70 units were applied respectively for N-30, N-60, and N-100 treatments). The experiment was conducted under rain-fed conditions for all experimental sites without irrigation and chemical weeding management (manual weed control). Cropping systems were sown on the 15th December in S1 and S3 sites, and on the 5th December in S2 site.

## 2.3. Sampling, Soil and Plant Measurements

Plant and soil variables were sampled at different periods from cropping cycle. In S1 and S3 sites, the soil was sampled 3 times: sowing, at 135 days (flowering) after sowing (DAS) and 180 DAS (harvest), while plant parameters were measured only during flowering and harvest stage. In S2 site, soil and plant were also sampled during the same periods, in which flowering and harvest stages were noted respectively at 121 and 167 DAS.

In case of soil analysis, the physico-chemical soil analysis in each experimental site was performed by using standard methods. Soil and plant total N content was determined by the Kjeldahl method [27]. However, the soil N-available concentration (N-NO$_3{}^{-}$ + N-NH$_4{}^{+}$) was measured using Henriksen's method. Soil pH was measured in the soil suspension with deionized water (soil:water ratio = 1:2.5) by using a pH meter [28]. The soil organic matter was measured using the Anne method [29], while the proportion of soil calcium carbonate (CaCO$_3$) was calculated by measuring the CO$_2$ volume, according to the Horton and Newson method [30]. For plant measurements, plants for both species were harvest in each sub-plot (in monocropped and intercropped sub-plots) replicate (0.25 m$^2$ quadrat). Shoots were separated from root zone and then dried in the oven (60 °C for 72 h), to determine the shoot dry biomass (SDB) per unit of land area. For each sampled plant, the leaves were taken, scanned by a portable leaf area measuring device (CID Bio-Science CI-202, www.cid-inc.com). The leaf area index (LAI) was then

calculated as the ratio of crop leaf area to land area [31]. Indeed, the leaf temperature was measured by an infrared thermometer (2950 chlorophyll analyzer Field-Scout CM 1000). However, the chlorophyll content was measured by the SPAD instrument. At harvest, grain yield (at 13 to 15% moisture content) and its components were estimated by harvesting all plants from the quadrat of 1 m$^2$ in each sub-plot (four replicate by each treatment). The grain protein uptake by grain yield (kg ha$^{-1}$) was calculated through the conversion of N content in grain yield (kg ha$^{-1}$) by using the conversion factor (k) for chickpea (k = 6.25) and durum wheat (5.7) crops [32]. For intercropped plots, both total grain yield and biomass were calculated as mixed crop by summing both yield and biomass of the two intercropped species.

*2.4. Water and Nitrogen Use Efficiency Calculation*

In order to follow the vertical water dynamics under the different treatments, soil moisture (gravimetric moisture) was determined by using the oven-drying method. It was measured relative to soil depth in each experimental site. Soil moisture measurements were performed in each soil layer from 0 to 20 cm, while they were covered with a soil depth from 0–40 cm for both S1 and S3 sites and from 0–100 cm in S2 site. However, the volumetric water content ($\Theta$ in m$^3$ m$^{-3}$) was determined by multiplying the gravimetric moisture by bulk soil density. To assess the amount of water stored in a soil profile (S: Equations (1) and (2)), we integrated the water content profile $\Theta$ (Z) down to the Z depth. We used the trapezoid method [33] to calculate the water stock in the soil; in general, the water stock is obtained by the following formulas:

$$S = \int_0^z \Theta(z)dz \tag{1}$$

$$S = \Theta_1 Z_1 + \sum \left( (\Theta_i + \Theta_{i-1})/2 \right) \cdot Z_i - Z_{i-1} \tag{2}$$

The actual water use (WU) was determined using the water balance equation (Equations (3) and (4)) for each cropping (N-fertilizer × level treatment) in all studied sites. For WU calculation, we used the simplified water balance equation, in which drainage and irrigation (no applied irrigation in the three field experiments) are considered as negligible according to our experiment site conditions. As that, only precipitation (P), and change in stored soil water ($\Delta$S) are considered in the calculation of WU (Equation (4)).

$$P - WU \pm \Delta S = 0 \tag{3}$$

$$WU = P \pm \Delta S \tag{4}$$

However, water use efficiency (WUE) was calculated relative to both whole biomass and grain yield that were measured during harvest period. Thus, WUE was calculated as the ratio between either plant whole biomass (WUE$_{YB}$: Equation (5)) or grain yield (WUE$_{GY}$: Equation (6)). In intercropped sub-plot, WUE calculation is based on the mixed biomass and grain yield of both intercropped chickpea and durum wheat.

$$WUE_{YB} = Total\ crop\ biomass/WU \tag{5}$$

$$WUE_{GY} = Grain\ yield/WU \tag{6}$$

In parallel, the NUE was also assessed under the different cropping system *N-fertilizer level treatments, in each experiment site. In this study, we adopted the approach to NUE that defines NUE as the fraction of N fertilizer that is utilized and allocated to relative yield of N [34]. As that, NUE was calculated as the ratio between N uptake by grain yield and N-applied by trough fertilization (Equation (7)). In intercropped plots, we used the mixed N uptake by grain yield by summing both N-grain yield of both intercropped chickpea and durum wheat.

$$NUE = N\text{-}uptake\ GY/N\ fertilizer \tag{7}$$

### 2.5. Statistical Analysis

Before proceeding to the analyses of variance (ANOVA), data normality was tested for all collected data. A one-way ANOVA was performed to assess the significant difference between soil chemical and physical proprieties among the three studied sites. Moreover, the effects of cropping system (Crop-Syst), N-fertilizer level (N-level) and interaction Crop-Syst × N-level on WUE, NUE, protein content, biomass and grain yield were assessed by using two-way ANOVA. The same analysis was performed for both chlorophyll and crop temperature variables, but with considering the following level for cropping system factor: chickpea monocrop, wheat monocrop, intercropped chickpea and intercropped wheat. For all performed ANOVA, the significant level of all treatments effect were presented at the *p*-value = 0.05. The means were compared by the Tukey's test where measured variables are significantly affected by treatments. Statistical analysis was performed using Statistica 8 for Windows.

## 3. Results

### 3.1. Chlorophyll and Crop Temperature

The chlorophyll (Chl) content in the leaf of chickpea and durum wheat in the different treatments (crop-syst × N-level) are given in Table 2 for each experimental site. Data show that Chl content was significantly ($p \leq 0.05$) affected only by crop-syst and, under S1 and S2 sites conditions. For the durum wheat crop, intercropping increased (7.80%) significantly chl content under low N-application (N-30), while chl content was decreased by more than 8% in durum wheat leaf intercropped under moderate and high (N-60 and N-100, respectively) N-fertilization. However, no significant changes were observed in chl content among the different crop-syst × N-level treatments (Table 2). In the case of chickpea crop, we note the higher chl content in N-100 rate when chickpea was sole cropped, while no significant difference was found between both sole-cropped and intercropped chickpea under either N-30 and N-60 rates.

**Table 2.** Crop temperature and chlorophyll (Chl) content in chickpea and durum wheat cultivated in sole crop and intercropping, under different crop-N level treatments in the different studied sites.

| | | S1 | | S2 | | S3 | |
|---|---|---|---|---|---|---|---|
| **Crop** | **N-Level** | **Chlorophyll (SPAD)** | **Temperature (°C)** | **Chlorophyll (SPAD)** | **Temperature (°C)** | **Chlorophyll (SPAD)** | **Temperature (°C)** |
| Durum wheat | N-30 | 36.7 ± 0.9 [a] | 36.3 ± 0.3 [a] | 39.3 ± 5.2 [a] | 20.9 ± 0.6 [a] | 50.0 ± 0.1 [a] | 24.7 ± 0.7 [a] |
| | N-60 | 37.6 ± 3.10 [ab] | 36.5 ± 1.2 [a] | 37.2 ± 3.1 [a] | 21.8 ± 1.2 [a] | 52.0 ± 3.4 [a] | 25.3 ± 0.7 [a] |
| | N-100 | 38.7 ± 4.6 [ab] | 37.0 ± 0.6 [a] | 40.1 ± 6.3 [a] | 20.3 ± 1.1 [a] | 47.1 ± 4.6 [a] | 26.3 ± 0.4 [a] |
| Chickpea | N-30 | 54.5 ± 44.7 [ab] | 39.8 ± 1.1 [ab] | 49.5 ± 10.6 [a] | 23.1 ± 2.1 [ab] | 47.2 ± 2.1 [a] | 21.5 ± 1.1 [a] |
| | N-60 | 55.1 ± 16.2 [ab] | 38.9 ± 1.1 [ab] | 54.2 ± 2.5 [a] | 24.4 ± 5.7 [b] | 45.6 ± 5.6 [a] | 24.3 ± 0.6 [a] |
| | N-100 | 65.0 ± 3.1 [b] | 38.0 ± 0.7 [ab] | 41.4 ± 15.6 [a] | 22.4 ± 0.8 [ab] | 48.9 ± 1.3 [a] | 25.2 ± 0.2 [a] |
| Wheat intercrop | N-30 | 39.5 ± 4.2 [ab] | 38.4 ± 1.2 [ab] | 41.8 ± 0.3 [a] | 20.1 ± 0.4 [a] | 51.4 ± 8.4 [a] | 32.5 ± 12.7 [a] |
| | N-60 | 32.9 ± 1.7 [a] | 38.0 ± 2.4 [ab] | 39.8 ± 2.7 [a] | 19.6 ± 0.6 [a] | 48.3 ± 6.8 [a] | 21.6 ± 1.1 [a] |
| | N-100 | 35.6 ± 8.79 [a] | 41.0 ± 0.3 [b] | 47.5 ± 6.3 [a] | 19.7 ± 0.8 [a] | 49.8 ± 5.4 [a] | 26.6 ± 0.9 [a] |
| Chickpea intercrop | N-30 | 48.3 ± 11.4 [ab] | 38.4 ± 1.2 [ab] | 46.5 ± 3.1 [a] | 20.7 ± 1.5 [a] | 47.0 ± 1.6 [a] | 25.3 ± 0.7 [a] |
| | N-60 | 54.6 ± 17.1 [ab] | 38 ± 2.4 [ab] | 48.8 ± 12.1 [a] | 19.6 ± 0.6 [a] | 45.4 ± 5.4 [a] | 21.6 ± 1.1 [a] |
| | N-100 | 55.5 ± 11.6 [ab] | 41.0 ± 2 [b] | 54.9 ± 4.2 [a] | 19.7 ± 0.8 [a] | 43.7 ± 3.4 [a] | 25.6 ± 5.7 [a] |
| *p* value | Crop-syst | ≤0.001 | ≤0.001 | 0.01 | ≤0.001 | 0.21 | 0.36 |
| | N-Level | 0.53 | 0.03 | 0.81 | 0.06 | 0.76 | 0.18 |
| | Crop × N-level | 0.82 | 0.04 | 0.30 | 0.04 | 0.84 | 0.15 |

Data are means ± standard error of 4 replicates. Mean values labeled with the same letter were not significantly different at *p* < 0.05.

Regardless of crop temperature, crop-syst, N-level and interaction crop-syst × N-level significantly affected ($p \leq 0.05$) the crop temperature of durum wheat and chickpea in S1 and S2 sites, while no significant effect was observed on this variable under S3 conditions (Table 2). Durum wheat temperature was increased by 2.1, 2.5, and 4 °C (respectively in N-30, N-60, and N-100 doses) in intercropping as compared to their respective crops in monoculture under S1 conditions. In S2 site, no changes were observed in wheat temperatures among both intercrop and monocrop system. However, crop temperature in chickpea was significantly increased (+4 °C), by intercropping, particularly under high N-application in the S1 site. In contrast, it was decreased by 2.4, 4.8, and 2.7 °C, respectively, in N-30, N-60, and N-100 doses where chickpea was grown as intercropping in S2 site (Table 2).

### 3.2. Growth and Yield Responses

The mean values of leaf area index (LAI,) shoot total biomass and grain yield of chickpea and durum wheat are reported in Table 3. Regardless LAI results, the interaction of crop-syst and N-level showed a significant effect ($p \leq 0.01$) on LAI values in the three studied sites. In S1 site, the highest LAI (2.42) was observed in the intercropped plot from the mixed crop and under low N-application, while in the durum wheat monocrop the highest values in high N-application under S2 (LAI = 8.83) and S3 (LAI = 2.41) conditions was noted. In terms of biomass accumulation, the highest shoot biomass was observed in durum wheat monoculture for all crop-syst × N-level treatments among the three experiment sites, except in S2 site, where mixed crop increased significantly shoot biomass (in N-30 dose) by 29.45% as compared to their respective monocropped wheat (Table 3).

**Table 3.** Leaf area index (LAI), shoot dry biomass (SDB) and grain yield in chickpea, durum wheat and crop mixture under different crop-N level treatments in the different studied sites.

| Crop | N-Level | S1 LAI | S1 SDB (kg ha$^{-1}$) | S1 Grain Yield (kg ha$^{-1}$) | S2 LAI | S2 SDB (kg ha$^{-1}$) | S2 Grain Yield (kg ha$^{-1}$) | S3 LAI | S3 SDB (kg ha$^{-1}$) | S3 Grain Yield (kg ha$^{-1}$) |
|---|---|---|---|---|---|---|---|---|---|---|
| wheat | N-30 | 2.26 ± 0.5 [de] | 10,965 ± 1164 [bc] | 2680 ± 635 [bc] | 5.53 ± 0.7 [bc] | 9071 ± 1420 [ab] | 7274 ± 523 [d] | 2.16 ± 0.4 [b] | 16,697 ± 1047 [cd] | 4619 ± 242 [c] |
| | N-60 | 1.60 ± 0.2 [bcd] | 14,464 ± 1862 [c] | 4637 ± 820 [c] | 8.05 ± 1.2 [cd] | 12,062 ± 1944 [bc] | 7546 ± 33.8 [d] | 1.16 ± 0.2 [a] | 18,470 ± 3521 [d] | 3952 ± 284 [c] |
| | N-100 | 1.65 ± 0.1 [be] | 10,195 ± 2612 [bc] | 4248 ± 368 [c] | 8.83 ± 2.4 [d] | 12,448 ± 3707 [bc] | 7761 ± 609 [d] | 2.41 ± 0.4 [b] | 13,659 ± 580 [c] | 4810 ± 47.4 [c] |
| Chickpea | N-30 | 0.57 ± 0.04 [a] | 696 ± 152 [a] | 115 ± 22.1 [a] | 4.26 ± 0.3 [ab] | 5639 ± 909 [a] | 2229 ± 61 [a] | 1.28 ± 0.2 [a] | 859 ± 234 [a] | 507 ± 51.6 [a] |
| | N-60 | 0.87 ± 0.1 [ab] | 493 ± 65 [a] | 175 ± 7.3 [a] | 3.41 ± 0.3 [ab] | 11,640 ± 1303 [bc] | 3559 ± 194 [ac] | 0.75 ± 0.003 [a] | 947 ± 209 [a] | 489 ± 32.8 |
| | N-100 | 1.02 ± 0.05 [ab] | 528 ± 40.1 [a] | 179 ± 34.7 [a] | 1.51 ± 0.2 [a] | 16,385 ± 3973 [c] | 2826 ± 1287 [ab] | 0.92 ± 0.08 [a] | 1108 ± 259 [a] | 325 ± 52.1 [a] |
| Mixed crop | N-30 | 2.42 ± 0.3 [e] | 6073 ± 779 [ab] | 1049 ± 86 [ab] | 2.96 ± 0.1 [ab] | 11,742 ± 373 [bc] | 3407 ± 250 [ac] | 0.87 ± 0.1 [a] | 5649 ± 1086 [b] | 2363 ± 128 [b] |
| | N-60 | 1.39 ± 0.07 [bc] | 7913 ± 178 [b] | 1850 ± 49 [ab] | 4.38 ± 0.3 [b] | 11,851 ± 1029 [bc] | 4689 ± 185 [bc] | 0.88 ± 0.2 [a] | 7800 ± 2036 [b] | 2253 ± 52.6 [b] |
| | N-100 | 2.02 ± 0.3 [ce] | 8726 ± 1733 [bc] | 2087 ± 112 [b] | 4.92 ± 0.8 [b] | 8033 ± 306 [ab] | 4840 ± 208 [c] | 0.82 ± 0.04 [a] | 7631 ± 944 [b] | 2106 ± 407 [b] |
| *p* value | Crop-syst | ≤0.001 | ≤0.001 | ≤0.001 | ≤0.001 | ≤0.001 | ≤0.001 | ≤0.001 | ≤0.001 | ≤0.001 |
| | N-Level | 0.008 | 0.26 | 0.03 | 0.09 | 0.004 | 0.01 | ≤0.001 | 0.07 | 0.41 |
| | Crop × N-level | 0.003 | 0.24 | 0.15 | ≤0.001 | 0.74 | 0.43 | ≤0.001 | 0.02 | 0.25 |

Data are means ± standard error of 4 replicates. Mean values labeled with the same letter were not significantly different at $p < 0.05$.

For grain yield, both crop-syst and N-level significantly affected the grain yield of chickpea and durum wheat in the S1 and S2 sites, while grain yield was only affected by crop-syst under S3 conditions. In S1 and S2 sites, N-application positively affected the grain yield of the mixed crop among the three applied doses of N-fertilizer. This positive effect was more pounced in the S1 site (by 98% passing from N-30 to N-100 dose) than in the S2 site (by 42% passing from N-30 to N-100 dose).

### 3.3. Protein Content in Grain Yield and Nitrogen Use Efficiency

Table 4 show the variation of NUE and protein content in grain yield of both species grown under the conditions of all crop-syst × N-level treatments. ANOVA analysis showed that protein accumulation was significantly ($p \leq 0.001$) affected by crop-syst among the three experimental sites.

**Table 4.** Nitrogen use efficiency (NUE) calculated at harvest and protein content in grain yield of chickpea, durum wheat and crop mixture under different crop-N level treatments in the different studied sites.

| Crop | N-Level | S1 Protein (kg ha$^{-1}$) | S1 NUE (kg kg$^{-1}$) | S2 Protein (kg ha$^{-1}$) | S2 NUE (kg kg$^{-1}$) | S3 Protein (kg ha$^{-1}$) | S3 NUE (kg kg$^{-1}$) |
|---|---|---|---|---|---|---|---|
| Durum wheat | N-30 | 243 ± 96.6 [bc] | 3.57 ± 0.5 [c] | 620 ± 57 [ac] | 4.75 ± 0.7 [b] | 748 ± 140 [c] | 6.79 ± 1.4 [c] |
| | N-60 | 341 ± 43.5 [c] | 3.16 ± 0.3 [c] | 663 ± 9.6 [bcd] | 4.20 ± 1.3 [b] | 688 ± 280 [bc] | 3.63 ± 1.4 [b] |
| | N-100 | 339 ± 133 [c] | 0.90 ± 0.2 [ab] | 907 ± 61.2 [d] | 1.91 ± 0.07 [a] | 956 ± 260 [c] | 2.38 ± 0.3 [ab] |
| Chickpea | N-30 | 15.1 ± 2.9 [a] | 0.29 ± 0.06 [a] | 342 ± 84.3 [a] | 2.89 ± 0.5 [ab] | 34.3 ± 3.4 [a] | 0.51 ± 0.08 [a] |
| | N-60 | 28.8 ± 0.9 [a] | 0.11 ± 0.01 [a] | 826 ± 147 [cd] | 4.06 ± 0.5 [b] | 56.5 ± 4.01 [a] | 0.23 ± 0.01 [a] |
| | N-100 | 17.9 ± 4.5 [a] | 0.10 ± 0.03 [a] | 386 ± 99 [ab] | 1.63 ± 0.1 [a] | 29.3 ± 4.7 [a] | 0.12 ± 0.02 [a] |
| Mixed crop | N-30 | 105 ± 32.5 [ab] | 8.09 ± 1.7 [d] | 561 ± 88 [ac] | 14.2 ± 0.7 [d] | 159 ± 23.9 [a] | 6.82 ± 2.1 [c] |
| | N-60 | 121 ± 39.7 [ab] | 2.61 ± 0.2 [bc] | 700 ± 56.8 [cd] | 7.88 ± 0.7 [c] | 324 ± 5.5 [ab] | 4.64 ± 0.8 [bc] |
| | N-100 | 162 ± 35.5 [ab] | 2.44 ± 0.7 [bc] | 910 ± 67 [d] | 4.29 ± 0.2 [b] | 350 ± 73.8 [ab] | 2.38 ± 0.4 [ab] |
| *p* value | Crop-syst | ≤0.001 | ≤0.001 | ≤0.001 | ≤0.001 | ≤0.001 | ≤0.001 |
| | N-Level | 0.18 | ≤0.001 | 0.038 | ≤0.001 | 0.15 | ≤0.001 |
| | Crop × N-level | 0.59 | ≤0.001 | 0.70 | ≤0.001 | 0.30 | 0.01 |

Data are means ± standard error of 4 replicates. Mean values labeled with the same letter were not significantly different at $p < 0.05$.

The highest production of protein was observed in grain yield of sole cropped durum wheat in all sites and N-level conditions, except for that produced by mixed crop in S2 site under moderate and high application of N fertilizer. This leads to increase protein in the grain yield of the mixed crop by 37 kg ha$^{-1}$ as compared to their respective crops in sole cropped durum wheat (Table 4). Results show also that N-application gradually increased protein accumulation only in the grain yield of mixed crops where protein production was increased by 16 and 41 kg ha$^{-1}$ in the S1 site (passing from N-30 to N-60 and from N-60 to N-100, respectively), 139 and 210 kg ha$^{-1}$ in S2 site and 165 and 26 kg ha$^{-1}$ in S3 site. In the case of the durum wheat monoculture, there was also a gradual increase with N application, particularly in S2, while we observed that the highest protein yield was obtained with N-60 level for sole-cropped chickpea.

Data in Table 4 show also the fraction of fertilizer N that is utilized and allocated to relative yield N, which was presented by NUE values. The NUE was significantly affected ($p \leq 0.01$), by crop-syst, N-level and crop-syst × N-level treatments among the three studied sites. NUE was significantly increased by intercropping, where the highest values of NUE was noted in the mixed crop as compared to sole-cropped durum wheat and chickpea. Indeed, reducing the N-application dose leads to a gradual increase in NUE; this increase was more pronounced in intercropping durum wheat–chickpea. Because of this, NUE by mixed crop was increased by 5.65, 9.91 and 4.44 kg kg$^{-1}$ passing from N-30 to N-100, respectively, in S1, S2, and S3 sites (Table 4).

### 3.4. Water Use and Water Use Efficiency

Data relative to WU and WUE by both whole plant biomass (WUE$_{YB}$) and grain yield (WUE$_{GY}$) are given in Table 5 for all crop-syst × N-level combinations in S1, S2, and S3 sites. According to ANOVA, WU by monoculture and mixed crop in S1 and S3 sites was significantly ($p \leq 0.05$) affected by both N-level and crop-syst × N-level interaction, while it was affected ($p \leq 0.01$) by crop-syst and crop-syst × N-level interaction under S2 conditions. In the chickpea and durum wheat mixed crop, the highest WU was noted in the S1 and S2 sites under low and moderate N application, respectively. Thus, intercropping increased water consumption by 23 and 58 m$^3$ ha$^{-1}$ as compared, respectively, to chickpea

and durum wheat sole cropped under low N-application and S1 conditions. This increase was more pronounced in the S2 site, in which the WU of chickpea–durum wheat mixed crop was greater as that observed in chickpea (by +282 m$^3$ ha$^{-1}$) and durum wheat (by +405 m$^3$ ha$^{-1}$) in monoculture. However, the highest value of WU in the S3 site was found in the monoculture of durum wheat under moderate N application, while there was no significant change in WU among the three cropping system (Table 5).

**Table 5.** Water use (WU), water use efficiency for whole biomass (WUE$_{YB}$) and water use efficiency for grain yield (WUE$_{GY}$) in chickpea, durum wheat and crop mixture under different crop-N level treatments in the different studied sites.

| Crop | N-Level | S1 | | | S2 | | | S3 | | |
|---|---|---|---|---|---|---|---|---|---|---|
| | | WU (m$^3$ ha$^{-1}$) | WUE$_{GY}$ (kg m$^{-3}$) | WUE$_{YB}$ (kg m$^{-3}$) | WU (m$^3$ ha$^{-1}$) | WUE$_{GY}$ (kg m$^{-3}$) | WUE$_{YB}$ (kg m$^{-3}$) | WU (m$^3$ ha$^{-1}$) | WUE$_{GY}$ (kg m$^{-3}$) | WUE$_{YB}$ (kg m$^{-3}$) |
| Durum wheat | N-30 | 3197 ± 50.3 [c] | 0.84 ± 0.3 [bc] | 3.44 ± 0.7 [bc] | 5444 ± 130 [c] | 1.33 ± 0.2 [c] | 1.67 ± 0.3 [ab] | 3019 ± 25.3 [ab] | 1.53 ± 0.09 [cd] | 5.53 ± 0.3 [c] |
| | N-60 | 3107 ± 33.8 [bc] | 1.49 ± 0.4 [c] | 4.66 ± 0.9 [c] | 5347 ± 157 [bc] | 1.41 ± 0.04 [c] | 2.26 ± 0.4 [b] | 3054 ± 41.9 [b] | 1.29 ± 0.3 [c] | 6.04 ± 1.1 [c] |
| | N-100 | 3032 ± 110 [ab] | 1.41 ± 0.4 [c] | 3.41 ± 1.6 [bc] | 5224 ± 107 [ab] | 1.48 ± 0.1 [c] | 2.38 ± 0.6 [bc] | 2736 ± 45.8 [a] | 1.76 ± 0.04 [d] | 4.99 ± 0.1 [c] |
| Chickpea | N-30 | 3232 ± 36.5 [c] | 0.03 ± 0.01 [a] | 0.21 ± 0.08 [a] | 5114 ± 71.4 [a] | 0.44 ± 0.1 [a] | 1.10 ± 0.1 [a] | 2982 ± 17.6 [ab] | 0.17 ± 0.01 [a] | 0.28 ± 0.07 [a] |
| | N-60 | 3125 ± 26.4 [bc] | 0.05 ± 0.01 [a] | 0.16 ± 0.03 [a] | 5224 ± 15 [ab] | 0.69 ± 0.04 [ab] | 2.23 ± 0.2 [b] | 3005 ± 50.3 [ab] | 0.16 ± 0.01 [c] | 0.31 ± 0.06 [a] |
| | N-100 | 2907 ± 56 [a] | 0.06 ± 0.01 [a] | 0.18 ± 0.01 [a] | 5525 ± 71.4 [cd] | 0.51 ± 0.2 [ab] | 2.97 ± 0.7 [c] | 2735 ± 294 [a] | 0.12 ± 0.02 [a] | 0.40 ± 0.06 [a] |
| Mixed crop | N-30 | 3255 ± 28.7 [d] | 0.36 ± 0.08 [ab] | 1.87 ± 0.3 [ab] | 5509 ± 33.9 [cd] | 0.62 ± 0.06 [ab] | 2.13 ± 0.05 [b] | 3007 ± 76.2 [ab] | 0.91 ± 0.1 [bc] | 1.89 ± 0.4 [b] |
| | N-60 | 3048 ± 9.79 [ab] | 0.51 ± 0.08 [ab] | 2.60 ± 0.09 [bc] | 5629 ± 4.5 [d] | 0.89 ± 0.07 [b] | 2.10 ± 0.2 [b] | 3007 ± 27.1 [ab] | 0.78 ± 0.01 [b] | 2.60 ± 0.6 [b] |
| | N-100 | 3013 ± 41.4 [ab] | 0.69 ± 0.1 [ab] | 2.9 ± 0.9 [bc] | 5283 ± 54.6 [b] | 0.92 ± 0.05 [b] | 1.52 ± 0.04 [ab] | 3017 ± 35.4 [ab] | 0.69 ± 0.1 [b] | 2.53 ± 0.2 [b] |
| *p* value | Crop-syst | 0.67 | ≤0.001 | ≤0.001 | 0.003 | ≤0.001 | 0.53 | 0.13 | ≤0.001 | ≤0.001 |
| | N-Level | ≤0.001 | 0.01 | 0.23 | 0.48 | 0.007 | 0.004 | 0.002 | 0.16 | 0.17 |
| | Crop × N-level | 0.02 | 0.14 | 0.30 | ≤0.001 | 0.25 | 0.001 | 0.05 | 0.02 | 0.15 |

Data are means ± standard error of 4 replicates. Mean values labeled with the same letter were not significantly different at *p* < 0.05.

Moreover, WUE$_{GY}$ was affected significantly (*p* ≤ 0.001) by both crop-syst and N-level in S1 and S2 sites and by crop-syst and crop-syst × N-level interaction (*p* ≤ 0.05) in the S3 site.

Among the three cropping systems, durum wheat grown in monoculture was most efficient system in term of WUE$_{GY}$ in both sites and N-application rates. It was then followed by mixture crop, for which WUE$_{GY}$ noted the highest values under either moderate or high N application in both S1 and S2 sites. While the highest values of WUE$_{GY}$ in S3 site were observed in low N application. As compared to chickpea monoculture, intercropping increased WUE$_{GY}$ by more than 0.30 and 0.57 kg m$^{-3}$ in S1 and S3 sites over the three N-application doses, and by more than 0.18 kg m$^{-3}$ under S2 conditions. In terms of WUE$_{YB}$, the measured values were only affected significantly (*p* ≤ 0.001) by crop-syst treatment in S1 and S3 sites, while both N-level and crop-syst × N-level interaction affected significantly (*p* ≤ 0.01) WUE$_{YB}$ under S2 conditions. In S1 and S3 sites, WUE$_{YB}$ was varied with the same trend as that of WUE$_{GY}$ among the three cropping systems. However, WUE$_{YB}$ by mixed crop was significantly increased in S2 site by +0.46 and +1.03 kg m$^{-3}$, as compared respectively to sole cropped chickpea and durum wheat under low application of N fertilizer (Table 5).

## 4. Discussion

Our results indicated the positive interaction between N and water use by chickpea–durum wheat intercropping, as mixed crop increased its total biomass, LAI and protein accumulation in grain yield (Tables 3 and 4). This improvement of growth and yield components was globally observed under low and moderate N-soil inputs (available from natural soil N and added fertilizer) and in either sub-humid (S2 site) or semi-arid (S1 site) climate. Legume-cereal intercropping was supported as beneficial practice to improve aboveground biomass and grain yield of intercropped cereal [20]. Recent field researches demonstrated that growth and yield performance of intercropped maize and

durum wheat are directly linked to the efficient use of N resource from biological $N_2$ fixation by intercropped soybean and faba bean, respectively [13,24]. This N facilitation by intercropping legume was also indicated as the key processes in increasing protein accumulation in grain yield of both cereals-legumes mixed crop [8,35]. Our results are in line with these last findings, showing a clear intercropping performance in terms of growth and yield quality especially in low-N deficient soils, and more so under sub-humid (S2) than semi-arid (S1) conditions. Such improvements that are probably due to high $N_2$-fixation by intercropped chickpea was also associated with a significant increase of chlorophyll content (Table 2). This was particularly demonstrated in leaf of intercropped durum wheat under low N-application. Several studies have addressed the effect of high residual soil mineral N (Soil naturel N + N-fertilizer) in inhibiting legumes $N_2$ fixation and decreasing efficiency in use of rhizobial symbiosis [14,36,37]. The higher grain yield of chickpea grown as both monocrop and intercrop system in S2 was probably related to favorable conditions for chickpea nodulation, as compared to that demonstrated in S1 (i.e., low water availability) and S3 (high available soil mineral N) experiment sites. Thus, low N soil availability and optimal water availability from rain leads to increase the efficiency in use of rhizobia symbiosis for chickpea grown under S2 conditions. Recent research studies on legumes-nodule diagnosis reported that water deficit and high N-available in initial soil often inhibits $N_2$ fixation by either sole cropped or intercropped legumes [16,20,36].

Our study, therefore, highlights an increase of N-fertilizer use by chickpea–durum wheat intercropping among the different pedoclimatic conditions, owing to higher NUE as compared to both sole cropped chickpea and durum wheat (Table 4). In terms of NUE, our results are consistent with these focused on maize-soya intercropping, which reported an increase of NUE by intercropping with more than 44.2% as compared to maize monoculture [38]. However, for forage intercropped species, previous research study demonstrated a significant decrease of NUE by intercropped oat-pea within 46 and 95% as compared to oat and pea monoculture, respectively [39]. Moreover, NUE improvement was demonstrated under either moderate or low N-application in semi-arid climate, while it was shown under all N-application doses under S2 conditions. Overall, a greater effect of intercropping on improving NUE was underlined under sub-humid climate as compared to that observed in semi-arid sites. This was probably due to much water availability in S2 site, by which excessive N-fertilizer was efficiently optimized during growth and yield development. Likewise, N demand by crop from N fertilizers application was globally reduced as consequence of a significant decrease in growth rate due to water scarcity, in particular under low rain-fed conditions [16,40]. As compared to chickpea monoculture, NUE was strongly enhanced under low N-application in semiarid conditions (S1 and S3) by intercropping. This may lead to mitigate water deficit effect thanks to better-optimization of N-fertilizers use when chickpea grown together with durum wheat.

In parallel, optimizing N use by mixed chickpea–durum wheat was simultaneously associated by increasing water use, in particular under low and moderate N application in S1 and S2 sites. This leads to greater WUE by aboveground biomass of mixed crop as compared to monoculture, but this was only observed in sub-humid conditions and under low-N application (Table 5). Nevertheless, durum wheat monoculture was the most efficient system in terms of water use for grain yield production as compared to mixed crop. This was probably due to higher grain yield of durum wheat in monoculture (as compared to intercropping), and which leads to increase WUE as consequence of increasing grain yield/WU ratio. The decrease in grain yield of legumes-cereals mixed crop in intercropping was presumably due to low-seeding density and interspecific competition between intercropped species [14,41]. These findings are similar with our results on chickpea–durum wheat intercropping particularly in semi-arid conditions. Regardless sub-humid conditions, intercropping increased biomass and protein accumulation in grain yield as compared to durum wheat monoculture, in particular under moderate N-application.

As compared to chickpea monoculture, the simultaneous assessment of NUE and WUE in this study showed a positive interaction between water and N use by mixed chickpea–durum wheat for grain yield production. This interaction leads to optimize N and water use by intercropped species over both contrasted climates and N-application levels. Recent studies have addressed separately the effect of intercropping on either NUE [16,21] or WUE [18]. According to these researches, intercropping system can increase NUE and WUE for both intercropped cereals and legumes; it was also identified as an efficient practice for enhancing of agroecosystem performance and resilience by reducing the requirement for synthetic N-fertilizers. Our results are in line with these reported on monoculture and which confirmed that water availability among both rainfall and irrigation greatly affects N status at soil and crop level [26]. As that, water limitation in semi-arid regions (S1 and S3 sites) leads to decrease drastically NUE by sole cropped wheat and chickpea where N application are not optimized according to the crop demand. However, intercropping promote the most efficient use of water by the two species under a wide range of climate conditions, this was simultaneously contributed to increase NUE, particularly under low and moderate N-application. In the same context, the increase in mixed crop temperatures (Table 2) under semiarid climate which was probably due to higher light interception, which leads to decrease soil evapotranspiration as consequence of reducing soil temperature. Legumes-cereals intercropping with fertilizer application reduced exposure and soil temperature, and increased soil water use by limiting soil-crop evapotranspiration [42].

Though interactions between water and N use in crop growth and yields are known for long time, while there are poorly studied in intercropping system [43], this leads to an ambiguous understanding of N and water management under this innovative cropping practice. The major innovative findings in this research paper demonstrate that fertilizer N applications in intercropping may stimulate growth and yield quality but may also lead to exhaustion of soil water in semi-arid climate and thereby to a relatively low grain yield when increasing N-application. Hence, both positive and negative interactions may occur between water and N use in chickpea–durum wheat intercrops yields, and in NUE and WUE, depending on climate and N mineral status in the soil.

## 5. Conclusions

The most of results demonstrated in this field investigation on simultaneous assessment of WUE and NUE by intercropped species were partially in line with recent findings reported in monoculture. The principal novelty found in this study concerned the positive interaction between the use of water and N by intercropped chickpea and durum wheat, as we confirmed that WUE and NUE were simultaneously increased in low and moderate N-application. This was particularly supported under both semi-arid and sub-humid climate conditions where intercropping was practiced in low N-soils availability. The simultaneous increase of WUE and NUE in wheat–chickpea intercropping was associated by a significant increase of both leaf-chlorophyll content in intercropped durum wheat under low N-application and an increase of mixed crop LAI and biomass, particularly under low N-application and sub-humid conditions. In addition, the increase of N-application from N-30 to N-100 leads to affect positively grain yield only for mixed crop with more increase under sufficient water availability (S2 site). N-supply via fertilization enhanced progressively protein accumulation only in grain yield of mixed crop under either low (S1 and S2 sites) or high (S3 site) N-soils. This increase in protein content was more pronounced in sub-humid climate as compared to semi-arid climate. Our results showed also that low N-application boosted intercropping to increase NUE under any condition of N-soil status (i.e., sufficient and deficient N-soil) and climate (i.e., sub-humid and semi-arid). This was accompanied by a simultaneous increase of WUE by chickpea–durum wheat mixed biomass, in particular under optimal water and low N-soil availability conditions (S2 site). Rain-fed chickpea–durum wheat intercropping is more advantageous in terms of growth

and yield quality as compared to monoculture; this may result from efficient optimization of water and N-fertilizer inputs in low N-soil and moderate water availability.

**Author Contributions:** F.-Z.B.: Manuscript writing, data collection, plant and soil sampling and laboratory analysis. S.H. and N.-Y.R.: Statistical analysis and general lecture and revision of manuscript sentences. O.K., B.Z., F.G., A.M. and S.A.: Contributions partially in data collection and plant and soil analysis. S.H.: Contribution in manuscript writing, and revision. M.L.: Methodology formulation, the manuscript writing, revision, and supervision, field management, and data collection and corresponding author. B.H.: Validation and Data curation. L.M.: Data curation. All authors have read and agreed to the published version of the manuscript.

**Funding:** This research received the financial funding by PRIMA (Grant Agreement no. 1912), a program supported by the European Union, research project "Research-based participatory approaches for adopting Conservation Agriculture in the Mediterranean Area–CAMA", Algerian coordinator Mourad Latati. This work was also supported by the PRFU project (D04N01ES160320190001) run by the Algerian Ministry of Higher Education and Scientific Research (M. Latati., Research work planification). This paper has been supported by the RUDN University Strategic Academic Leadership Program.

**Conflicts of Interest:** The authors declare no conflict of interest.

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
