# Peer review of "The Effect of Nitrogen Supply on Water and Nitrogen Use Efficiency by Wheat–Chickpea Intercropping System under Rain-Fed Mediterranean Conditions"

_agriculture, doi:10.3390/agriculture13020338_

Round 1

Reviewer 1 Report

Dear Authors,

The submitted manuscript addresses a very important and topical topic of N and water use in crops. Changing climate causing very often water shortages for plants significantly affects crop yields and quality. In addition, mineral N fertilisation very often represents a significant share of farmers' costs, so it is important to assess its use.

In my opinion, the manuscript needs many changes and clarifications.

Line 56: Many studies show that leguminous crops have a higher weed pressure, so their introduction does not suppress weeds. I think this sentence is incorrectly phrased. Indeed, mixed cropping does reduce weed pressure, but in relation to legumes and not cereals

Line 65 - 68: this is a repetition from earlier lines

Line 74: please provide the source of the estimate

Line 94: citation form error

Line 176: why are the contents of other important minerals such as P and K not shown?

Line 191: please explain why this blend composition was chosen?

Line 195: no chemical weed control was used or none at all? As I mentioned legumes are susceptible to weeds, in addition high N fertilisation rates promote weed growth so both the pure chickpea crop and the mixture may have exhibited significant weed infestation which may have translated into yields of both biomass, grains and protein

In general, this section should be completed with information on the form of N fertilisation applied and over what period. Was N fertilisation applied as a single dose or was it split? Was any other mineral fertilisation with P and K applied?

Line 213: an area of 0.25m2 I think is too small for a representative sampling of the plants especially in mixtures I suggest in future studies on mixtures to use a square of 1m2

Line 214: I understand that the result obtained was referred to as biomass in the tables? If so please clarify the description.

Line 222 - 223: content is usually given in % or g/kg dm. If it was converted from kg/ha-1 then it was converted from yield/uptake not content. I think this is a language issue please verify this. Earlier it was stated how the N content was determined, there is no explanation of how the N yield in kg/ha-1 was determined

Line 226: How was the ratio of plants at harvest. Were the ratios appropriate as to the seed sown? Many studies report strong competitive interactions in legume-cereal mixtures. In general, you get lower ratios at harvest relative to seeding. An additional factor here may be nitrogen fertilisation which increases the competitiveness of cereals against legumes. I think this may have had an effect on the yields obtained especially protein in the biomass.  The authors mention this in the literature cited later in the paper, but do not state whether this was similarly the case in the authors' study

Line 293 - 303: the authors report crop temperature results but this is not covered in any way in the discussion section, so for what purpose were these results reported?

Line 333: wrong table number given

Line 336: In the case of wheat at sites S1 and S2 there was also a gradual increase with N application, ambiguously only obtained at S3. In legume, a trend can also be observed, in all objects the highest protein yield was obtained with N60 fertilisation. Please consider this

Line 388: Figures? I suspect the authors had tables in mind

Line 400: In the results section the authors stated that the highest protein yield was obtained from the wheat crop. Table 4 shows that there was only a slight improvement in protein yield at site S2 (between the wheat and mixture crop), in terms of biomass the results were only similar at site S2. Please analyse this. I think it would also be good to consider a comparison between objects S1, S2, S3 where especially in object S2 higher yields of legume and mixture were obtained. What were the reasons for this?

Line 405: This is a very important issue when considering N fertilisation in mixtures and legume crops, is this reflected in the authors' bans?

Line 411: citation form error

Line 419: What does the high level of NUE indicate? For example, at site S2 in the mix at N30 there was a NUE of 14.2 so 14 times more N was taken up by the crop compared to fertiliser. The soil N content was much lower than at S3 where, by comparison, the NUE was more than 50% less. In contrast, the chickpea crop obtained an NUE value of less than 1 on all sites. I believe that this fact should be included in the discussion. Why such differences were obtained, why such a level of NUE was obtained and what this might lead to. Additionally, the unit kg kg-1 seems to me not commonly used much more often the expression % or dimensionless is encountered.

Line 431: So why wasn't a higher density used?

Line 445: Why is there such a correlation, the authors believe?

Line 451: Intercropping is not a novel approach to growing crops

Additionally:

I am puzzled as to why the results presented chose to present protein yield and not N yield since the paper is about N use? I think the presentation of N yield was much better. I also think that the paper should have been supplemented with N content data (% or g/kg dm). The authors mention in the materials and methods section the uptake of the material in two phases, and present the water use in both harvest phases, why was this not also done for N use? Legume-cereal mixtures are also used as green fodder for animals and thus harvested at the flowering stage. Is it not a common method to determine NUE for the different growth phases of the plants?

I also believe that the introduction should be supplemented by information on the impact of a deficiency or optimum availability of N and water for legumes, cereals and their mixed cultivation.

There are numerous inconsistencies in the reference section as regards form. In addition, the names of the authors are incorrectly stated in item 35 (problem with special characters related to the language of the country). This section must necessarily be corrected

Author Response

The Effect of Nitrogen Supply and Water Availability on Water and Nitrogen Use Efficiency by Wheat-Chickpea Intercropping System under Rain-fed Mediterranean Conditions

Fatima-Zohra Bouras1, Salah Hadjout2, Benalia Haddad1, Asma Malek1, Sonia Aitmoumen1, Feriel Gueboub1, Luiza Metrah1, Bahia Zemmouri1, Omar Kherif1, Nazih-Yacer Rebouh3 and Mourad Latati1,3,*

1   Ecole Nationale Supérieure Agronomique (ES1603), Laboratoire d’Amélioration Intégrative des Productions Végétales (C2711100), Département de Productions Végétales, Avenue Hassane Badi, El Harrach, Algiers 16200, Algeria; bourasfatmazohra@yahoo.fr (F.Z.B); haddadbenalia@yahoo.fr (B.H); bahia.zemm@gmail.com (B.Z); kherifomar@gmail.com (O.K.); m.latati@yahoo.com (M.L)

2   Centre de Recherche en Aménagement du Territoire, CRAT, Campus Universitaire Zouaghi Slimane, Constantine, Algérie; hadjout.salah@gmail.com

3   Department of Agro-Biotechnology, Institute of Agriculture, Peoples’ Friendship University of Russia (RUDN University), 83843 Moscow, Russia; n.yacer16@outlook.fr (N.Y.R)

Correspondence: m.latati@yahoo.com; Tel: +213 671606269ï¼›Fax: +213 21822729

Response to Reviewers comments for the manuscript ID: agriculture-2180414

Dear Pr. Elvira Sun. Field Editor in AGRICULTURE journal,

      We have received the reviewer’s feedbacks on our Manuscript No. agriculture-2180414The Effect of Nitrogen Supply and Water Availability on Water and Nitrogen Use Efficiency by Wheat-Chickpea Intercropping System under Rain-fed Mediterranean Conditions”

      We are very grateful to the reviewers for their time and constructive comments. Based on those comments, we have been able to incorporate changes to reflect all of the suggestions provided by the reviewers in order to improve our manuscript. The changes made have been highlighted in green within the manuscript.  Moreover, we proceeded to an English proofreading of the manuscript a carefully checked for errors.

     Please, also find below in green, our point-by-point responses the reviewers’ comments.

      We would like to thank Editor and all reviewers and the referees for your valuable time and efforts.

                                                        Dr. LATATI Mourad

Reviewer 1

Dear Authors,

The submitted manuscript addresses a very important and topical topic of N and water use in crops. Changing climate causing very often water shortages for plants significantly affects crop yields and quality. In addition, mineral N fertilisation very often represents a significant share of farmers' costs, so it is important to assess its use.In my opinion, the manuscript needs many changes and clarifications.

Comment 1 (C1): Line 56: Many studies show that leguminous crops have a higher weed pressure, so their introduction does not suppress weeds. I think this sentence is incorrectly phrased. Indeed, mixed cropping does reduce weed pressure, but in relation to legumes and not cereals

Response 1 (R1): Done. We agree with you dear reviewer. The sentence was rephrased in the text. The new reformulated sentence is: “In addition, legumes are introduced in mixture with cereal crops to reduce soil erosion, increase land and resources (i.e. water and nutrients) use efficiency”. Modification was reported in L-53-54.

C2: Line 65 - 68: this is a repetition from earlier lines

R2: Done. We agree with you dear reviewer. The sentence was rephrased and repetition was deleted. Please see in  L-63-65.

C3: Line 74: please provide the source of the estimate

R3: Done. We agree with you dear reviewer. The reference was added in the text. Please see in L-74. And L635 in reference list

Line 94: citation form error

R4: R3: Done. We agree with you dear reviewer. The reference was reformulated to better adapt the reference form. Please see L-95-97.

C4: Line 176: why are the contents of other important minerals such as P and K not shown?

R4: Because we focus in our study on N soil status and water availability among the three studied site, so dear reviewer we have not given more details for not focussing on soil fertility but we focussed on N-soil content and climate variability. Indeed, we underline that calcareous content and pH and N soil content were the most indicators that varied significantly among the three studied sites.

C5: Line 191: please explain why this blend composition was chosen?

R5: Done. We agree with you dear reviewer. Regardless our methodological approaches, our experiment were performed under farmers practices in terms of intercropping management. As that, the planting density was chosen according to local standard cultural practices that are commonly adopted by farmers. Please see L-189-190.

C6: Line 195: no chemical weed control was used or none at all? As I mentioned legumes are susceptible to weeds, in addition high N fertilisation rates promote weed growth so both the pure chickpea crop and the mixture may have exhibited significant weed infestation which may have translated into yields of both biomass, grains and protein

In general, this section should be completed with information on the form of N fertilisation applied and over what period. Was N fertilisation applied as a single dose or was it split? Was any other mineral fertilisation with P and K applied?

R6: 1.Yes dear reviewer we have not applied any weed treatment, all weeds are removed from plots manually, this manual weeding was adopted to better (i) ensure that no interaction effect will occur between pesticide product and N-fertilizer, (ii) this also related to the local farmers practices that manage intercropping system without pesticide inputs for sustainable production and (iii) the most of pesticide products have not the co-effect on both chickpea and wheat intercropped species, globally they are selective only for only one species.

  1. Yes off-course we applied only N-fertilization as we mentioned in the text. However, we agree with you in terms of explanation of N-application timing and the type of fertilizer. The explanation was reported in the text about these amendments. Thus, N-application was applied with urea (46-0-0) during two periods from cropping cycle. The first N-supply was applied at beginning of durum wheat tillering, while the second one was applied at beginning of stem elongation. Please see L194-197

C7: Line 213: an area of 0.25m2 I think is too small for a representative sampling of the plants especially in mixtures I suggest in future studies on mixtures to use a square of 1m2

R7: Yes, I agree with you and I just underline that we used 1 m2 for grain yield estimation, however the use of 0.25m2 for another parameters in our study was supported by 4 sub-plot replicates and also this may provide to justify our experimental approach that is globally based on controlled. However we will pass to 1 m2 in our next research. Thanks for your recommendation.

C8: Line 214: I understand that the result obtained was referred to as biomass in the tables? If so please clarify the description.

R8: Done. We agree with you dear reviewer. Clarification was reported in the text. Please see L217

C9: Line 222 - 223: content is usually given in % or g/kg dm. If it was converted from kg/ha-1 then it was converted from yield/uptake not content. I think this is a language issue please verify this. Earlier it was stated how the N content was determined; there is no explanation of how the N yield in kg/ha-1 was determined.

R9: Done. Yes I agree with you, we rephrased this sentence to “The grain protein uptake by grain yield». For given you our reasoning in this context, the first estimation of protein was determined by % however then we converted in kg ha (relative to grain yield per ha) to better represent the protein accumulation by land area. The N dosage method was mentioned in the text: “Soil and plant total N content was determined by the Kjeldahl method through mineralization and distillation method [27]”. However I underline that logically we measured N grain Yield and which was used to calculate NUE, while results were not represented because we prefer to represent N protein which is considered as key indicator of yield quality. L 225

C10: Line 226: How was the ratio of plants at harvest. Were the ratios appropriate as to the seed sown? Many studies report strong competitive interactions in legume-cereal mixtures. In general, you get lower ratios at harvest relative to seeding. An additional factor here may be nitrogen fertilisation which increases the competitiveness of cereals against legumes. I think this may have had an effect on the yields obtained especially protein in the biomass.  The authors mention this in the literature cited later in the paper, but do not state whether this was similarly the case in the authors' study.

R10: Done.Yes we agree with you, as that we give the state and comparison of our finding with another studies in this context. This was added to discussion section. Please see L453-457 Moreover, we Off-course we agree with you. The ratio of plant was lower than that computed during the initial seeding in all cropping system and not only intercropping, and it was varied among the sites . The plant density was measured during emergence and harvest period to better estimate yield and also biomass. However, in intercropping the applied seeding dose were logically lower for both species as compared to sole cropping to reduce competitively between the two intercropped species. Yes we agree with you interspecific competition reduces yield and protein accumulation in biomass of mixed crop. We underline that we mentioned all these explanation in discussion section to better defend discussion section.

C11: Line 293 - 303: the authors report crop temperature results but this is not covered in any way in the discussion section, so for what purpose were these results reported?

R11: Done. Thanks for this important comment, I agree with you. We discussed the main results of crop temperature in discussion section. The modification (In the same context, the increase in mixed crop temperatures (Table 2) under semiarid climate which was probably due to higher light interception, which leads to decrease soil evapotranspiration as consequence of reducing soil temperature. Legumes-cereals intercropping with fertilizer application reduced exposure and soil temperature, and increased soil water use by limiting soil-crop evapotranspiration [43].) was added in the text. Please see L474-478

C12: Line 333: wrong table number given

R12: Done. The number of table was corrected in the text. Please see L-338.

C13: Line 336: In the case of wheat at sites S1 and S2 there was also a gradual increase with N application, ambiguously only obtained at S3. In legume, a trend can also be observed, in all objects the highest protein yield was obtained with N60 fertilisation. Please consider this

R13: Done. But here dear reviewer we agree your comment on gradual increase in protein only for wheat only in S2 because in S1 there are no difference between N-60 and N-100. So your proposition was rephrased to “In the case of durum wheat monoculture, there was also a gradual increase with N application particularly in S2, while we observed that the highest protein yield was obtained with N-60 fertilisation for sole cropped chickpea”. Please see L341-344

C14: Line 388: Figures? I suspect the authors had tables in mind

R14:Done. Thanks, yes offcourse we mean table 3 and 4 and not figure. Please see L397

C15: Line 400: In the results section the authors stated that the highest protein yield was obtained from the wheat crop. Table 4 shows that there was only a slight improvement in protein yield at site S2 (between the wheat and mixture crop), in terms of biomass the results were only similar at site S2. Please analyse this. I think it would also be good to consider a comparison between objects S1, S2, S3 where especially in object S2 higher yields of legume and mixture were obtained. What were the reasons for this?

R15: Done. Yes I agree with your, this is a crucial and key question. As that, In S3, soil was characterized by higher N-availability that affect negatively (reduce or inhibe) N2 fixation by chickpea, while the S1 condition are more favourable in terms of N soil availability (low to moderate N-soil availability) by water availability was not much adequate for chickpea growth. Conversely, in S2, both N soil availability and water supply from rain are favourable to nodule growth and N2 fixation. As your recommendation we proposed this sentence to better improve this discussion section:”The higher grain yield of chickpea grown as both monocrop and intercrop system in S2 was probably related to favorable conditions for chickpea nodulation, as compared to that demonstrated in S1 (i.e. low water availability) and S3 (high available soil mineral N) experiment sites. Thus, low N soil availability and optimal water availability from rain leads to increase the efficiency in use of rhizobia symbiosis for chickpea grown under S2 conditions. Recent research studies on legumes-nodule diagnosis reported that water deficit and high N-available in initial soil often inhibits N2 fixation by either sole cropped or intercropped legumes [16,20,37].” Please see L414-422

C16: Line 405: This is a very important issue when considering N fertilisation in mixtures and legume crops, is this reflected in the authors' bans?

R16: Yes offcourse dear reviewer, this was well reflected.

C17: Line 411: citation form error

R17: Done. The reference was corrected by reforming their integration in the text. Please see L428

C18: Line 419: What does the high level of NUE indicate? For example, at site S2 in the mix at N30 there was a NUE of 14.2 so 14 times more N was taken up by the crop compared to fertiliser. The soil N content was much lower than at S3 where, by comparison, the NUE was more than 50% less. In contrast, the chickpea crop obtained an NUE value of less than 1 on all sites. I believe that this fact should be included in the discussion. Why such differences were obtained, why such a level of NUE was obtained and what this might lead to. Additionally, the unit kg kg-1 seems to me not commonly used much more often the expression % or dimensionless is encountered.

R18: Done. First for the unit kg kg-1 it was the adequate unit according to our NUE calculation and which mean kg of N uptake relatively to 1 kg of N-fraction of N-fertilizer, we want to state this unit to better compare NUE with WUE changes. Second, for NUE discussion, I just want to underline you dear reviewer that NUE values were not less than 1 in all sites for chickpea, it was less than 1 only in S1 and S3 and this was due to lower rate of growth and nodulation in these two site as consequence of water scarcity in this sites. This low NUE was improved by intercropping effect particularly under low and moderate N-application. To better improve this discussion section we added your recommendation by the following explanation “As compared to chickpea monoculture, NUE was springily enhanced under low N-application in semiarid conditions (S1 and S3) by intercropping. This may lead to mitigate water deficit effect thanks to better-optimization of N-fertilizers use when chickpea grown together with durum wheat.” Please see L438-442

C19: Line 431: So why wasn't a higher density used?

R19: This mean that when we adopt intercropping we need reduce both species density to maintain crops interspecific competition (we cannot use higher density in intercropping because we cannot seem both intercropped species within the same density as that in monoculture). May be in the next future research we will try to increase the density of both intercropped species by not much or not equal to that of monoculture

C20: Line 445: Why is there such a correlation, the authors believe?

R20: Because water limitation allow to decrease relatively (as compared to optimal water availability) growth parameters of sole cropped specie, so the absorbed or available N fertilizer was not efficiently used by crop as consequence of reducing growth and yield processes in the absence of sufficient water supply.

C21: I am puzzled as to why the results presented chose to present protein yield and not N yield since the paper is about N use? I think the presentation of N yield was much better. I also think that the paper should have been supplemented with N content data (% or g/kg dm). The authors mention in the materials and methods section the uptake of the material in two phases, and present the water use in both harvest phases, why was this not also done for N use? Legume-cereal mixtures are also used as green fodder for animals and thus harvested at the flowering stage. Is it not a common method to determine NUE for the different growth phases of the plants?

R21: First, Yes offcourse Legume-cereal mixtures are also used as green fodder for animals and thus harvested at the flowering stage, however this not a case for chickpea crop that is cultivated for human food nutrition. We do not represent N flowering uptake because we focussed on grain yield which represent the first interest for cropping wheat and chickpea. I underline that logically we measured N grain Yield and also N in biomass and which were used to calculate NUE, while results were not represented because we prefer to represent N protein which is considered as key indicator of yield quality. I addition when we mentioned NUE it was clearly that the change of NU is correlated with change in N grain yield. We are ok for adding our data in terms of N grain yield as supplementary material.

C22: I also believe that the introduction should be supplemented by information on the impact of a deficiency or optimum availability of N and water for legumes, cereals and their mixed cultivation.

R22: Done. We added this recommendation in introduction section. Please see L65-68

C23: There are numerous inconsistencies in the reference section as regards form. In addition, the names of the authors are incorrectly stated in item 35 (problem with special characters related to the language of the country). This section must necessarily be corrected

R23: Done. All references section was checked. Modifications were reported in list of reference. Please see the new checked list of references.

Reviewer 2 Report

Specific comments

As a researcher in the field of farming, I am very interested in your work. The paper as a whole was informative and involved two years of experiments at three sites, under three different levels of nitrogen application and two different climatic conditions, yielding meaningful results. I have looked thoroughly at your article and I see that you did a lot of work on it.

However, There are some problems in the article that need to be solved, if I understand your description correctly, for instance, the growth and yield of the two intercropped crops were not analyzed and discussed separately. For the abstract and summary sections, there is too much content and not enough focus. As far as I see, the paper can be accepted if all the points are dealt with appropriately.

Abstract

1. Line 16: the summary section is comprehensive, but too long and unfocused.

2. Line 33, the article does not give specific data to confirm the significant improvement in NUE under intercropping conditions.

Introduction

3. Line 82,e.g. N, phosphorus and calciumshould be changed e.g. nitrogen, phosphorus and calciumor e.g. N,P,Ca.

4. Line 94: is the citation format for "Chen et al. (2018)" correct?

5. Line 109: is the font of "VIA" correct?

6. Line 109, "via" is italicized.

7. Line 109,Some font formats are inconsistent. Please check for modification.

8. Lines 123–132: whether the research objectives can be made into hypotheses.

9. In Table 1 , the unit bracket format is inconsistent with the other figures.

10. In Table 1,p valueshould be written in a line.

11. In Table1, the table format is different from other tables.

Materials and Methods

12. Lines 138–139: the format of the geographic coordinates is not correct.

13. Lines 137 and 138, latitude and longitude are incorrectly expressed.

14. Line137-138, latitude and longitude are incorrectly expressed.

15. Line 137,Algiers (36â—¦420 N, 3â—¦090 E)should be changedAlgiers (36°420 N, 3°090 E).

16. Line 138,it is the same as line 137,longitude and latitude are expressed incorrectly.

17. In Figure1, shorthand for temperature is not Tmoy.

18. Line 188, misspelled words,desighnshould readdesign.

19. Line 191,was150should readwas 150.

20. Line 202,harvest stage wereshould readharvest stages were 

21. Line 223,ha-1should readha-1.

22. Equation 4 should read WU= P±ΔS.

23. Figure 1,there is a missing icon here.

Results

24. Line 276: Much of the data for wheat and chickpea in intercropping were analyzed together, failing to separate the effects of intercropping on a particular crop and failing to capture the differences between a particular crop in monocrop and intercrop patterns.

25. Line 289: the "SPAD" in "Chlorophll SPAD" in Table 2 is missing a parenthesis and the data analysis of the lack of significance of temperature in Chickpea intercrop at the N-100 level (S1).

26. Line 291-292, the statement is ambiguous and it is recommended to modify it.

27. Line 294,canopy temperature is not reflected in Table2Line 331,wasobserved” should read was observed.

28. Line 311,wasobservedshould read was observed.

29. Line 331, treatment under N-100 also does not conform to general rules.

30. Line 333: here it should be Table 4, not Table 3.

31. Line 344: is the "+" in "+5,65, +9.91" necessary?

32. In Table 1 and 2, values are the mean ± error standard” does not appear,please add.

33. In Table 3 and 5,the arrangement of the data is different from that in Table 2 and 4.

34. Line 346-347, incorrect use of punctuation.

35. In Table1 and Table2, there is no description values are the mean ± error standard.

36. Table 2,Wheat intercrop and Chickpea intercrop What exactly does that mean? Arent they in conflict? Because there are only three planting factors in 2.2.

37. Table 4,0,01The comma here should be replaced by the decimal point.

38. Table 5,line 367,water use efficiency for whole biomass (WUEbh) and water use efficiency for grain yield (WUEgy) in chickpeashould be changed water use efficiency for whole biomass (WUEYB) and water use efficiency for grain yield (WUEGY) in chickpea

39. Table 5,3107± 33.8bc should be modified 3107± 33.8bc

Discussion

40. Line 388,(Figs. 3 and 4)should be modified (Table.3 and 4).

41. Line 390,sem-iarid (S3 site) climateshould be changed semi-arid (S3 site) climate .

42. Lines 400–402: is there any basis for the inference that "Such improvements that are probably due to a higher ability of intercropped chickpea to increase N2-fixation was also associated with a significant increase of chlorophyll content (Table 2) only in leaf of intercropped"?

43. Line 443, there is a space before the new sentence.

Conclusion

44. Line 458: the conclusion was written comprehensively, but there was too much content and the main conclusions were not highlighted enough.

References

45. There are numerous problems with the format of the references, such as wrong symbols in the authors, italicized issue numbers and journal names, etc.

46. Line 525, there are several errors in the references, for example, the year of the research is not bold, etc.

Author Response

Reviewer 2

Specific comments

As a researcher in the field of farming, I am very interested in your work. The paper as a whole was informative and involved two years of experiments at three sites, under three different levels of nitrogen application and two different climatic conditions, yielding meaningful results. I have looked thoroughly at your article and I see that you did a lot of work on it.

C1: However, There are some problems in the article that need to be solved, if I understand your description correctly, for instance, the growth and yield of the two intercropped crops were not analyzed and discussed separately. For the abstract and summary sections, there is too much content and not enough focus. As far as I see, the paper can be accepted if all the points are dealt with appropriately.

R1: Thanks very much for your recommendation. We applied all your comment and correct all your observed correction. Please see in below the response on all of your comments.

Abstract

C2: Line 16: the summary section is comprehensive, but too long and unfocused.

R2: Done. We agree with you dear reviewer, as that we shorted our abstract deeply to better give the lector clear idea. Please see the modification that reported in abstract section. See L 16-18, L20, L21-24 and L38-40.

C3: Line 33, the article does not give specific data to confirm the significant improvement in NUE under intercropping conditions.

R3: Specific data are available in our results since we evaluated and compared NUE amon all cropping systems (wheat sole crop, chickpea sole crop and intercropping) as showed by table4. So regarding to our results reported in table 4 in which we give all values of NUE for chickpea, Durum wheat and crop mixture under different crop and N-fertilizer  level treatments. So, the difference between intercropping (mixed crop) and monoculture for both species was statistically studied by comparing NUE in the same unit of land area….for this we compared mixture NUE with monoculture. We cannot separate the co-use of N-fertilizer by intercropped species that share the same space, so we calculated the co-use efficiency by summing both N uptake by intercropped grain yield and which was compared with monoculture in the same unit of land area.   

Introduction

C4:Line 82,“e.g. N, phosphorus and calcium”should be changed “e.g. nitrogen, phosphorus and calcium”or “e.g. N,P,Ca”.

R4: Done. Modification was reported in the text. Please see L82-83

C5:Line 94: is the citation format for "Chen et al. (2018)" correct?

R5: Done. Offcourse we need to put number reference. Thanks. Modification was reported in the text. Please see L 95-97

C6: Line 109: is the font of "VIA" correct? via" is italicized?.

R6: Yes dear reviewer because is considered as LATIN word

C7: Line 109,Some font formats are inconsistent. Please check for modification.

R7: Done. All font formats are checked. Modification was reported in the text. Please see L 110

C8: Lines 123–132: whether the research objectives can be made into hypotheses.

R8: Dear reviewer, as given both main and specific objectives, the specific objectives was performed by question formulation which can help lector to identify our hypothesis (response Yes or now). We give also the response on our question hypothesis in discussion section.

C9: In Table 1, the unit bracket format is inconsistent with the other figures.

R9: Done. Table 1 format was checked and newly formulated in terms of format. Please see table 1.

C10: In Table 1,“p value”should be written in a line.

R 10: Done. P value was well fitted with the new formulated table 1. Please see table 1.

C11: In Table1, the table format is different from other tables.

R11: Yes offcourse, so we transfer it in the same format of another tables. Please see table 1.  

Materials and Methods

C12: Lines 138–139: the format of the geographic coordinates is not correct.

R12: Done. All geographic coordinates were corrected in terms of format. Modification was reported in the text.  Please see L138-139.

C13: Lines 137 and 138, latitude and longitude are incorrectly expressed.

R13: Done. latitude and longitude were well-corrected . Modification was reported in the text.  Please see L138-139

C14: Line137-138, latitude and longitude are incorrectly expressed.

R14: Done. Please see our response 13

C15: Line 137,“Algiers (36â—¦420 N, 3â—¦090 E)”should be changed“Algiers (36°420 N, 3°090 E)”.

R15: Done. Here also we corrected not only degree but also number in minute position. Modification was reported in the text. Please see L1

38-139

C16: Line 138,it is the same as line 137,longitude and latitude are expressed incorrectly.

R16: Done. The expression of longitude and latitude was corrected for all studied sites. Modification was reported in the text. Please see L138-139

C17: In Figure1, shorthand for temperature is not “Tmoy“.

R17: Done. All figure compounet (legends, format…etc) were corrected. Please see Figure 1.

C18: Line 188, misspelled words,“desighn”should read“design”.

R18: Done. Modification was reported in the text. Please see L188

C19: Line 191,“was150”should read“was 150”.

R19: Done. Modification was reported in the text. Please see L191

C20: Line 202,“harvest stage were”should read“harvest stages were”

R20: Done. Modification was reported in the text. Please see L205

C21: Line 223,“ha-1”should read“ha-1“.

R21: Done. Modification was reported in the text. Please see L226

C22: Equation 4 should read “WU= P±ΔS“.

R22: Done. Modification was reported in the text. Please see equation 4

C23: Figure 1,there is a missing icon here.

R23: Done. Figure 1 was completely reformulated. All missing components were corrected. Please see figure 1.

Results

C24: Line 276: Much of the data for wheat and chickpea in intercropping were analyzed together, failing to separate the effects of intercropping on a particular crop and failing to capture the differences between a particular crop in monocrop and intercrop patterns.

R24: Dear reviewer. We separated in particular effect among both intercropped species only in chlorophyll and crop temperature. So why? Because chlorophyll content is specific for each crop species we cannot calculate the sum of both value in intercropping, it the same reason for crop temperature. Here we compared each monoculture species with their respective in intercropping.

C25: Line 289: the "SPAD" in "Chlorophll SPAD" in Table 2 is missing a parenthesis and the data analysis of the lack of significance of temperature in Chickpea intercrop at the N-100 level (S1).

R25: Done. Modification was reported in table 2 and homogeneous group was added. Please see Table 2.

C26: Line 291-292, the statement is ambiguous and it is recommended to modify it.

R26: Done. This sentence was rephrased to “Data are means ± standard error of 4 replicates. Mean values labeled with the same letter were not significantly different at p < 0.05. Modification was reported in the text. Please see L304-307.

C27: Line 294, canopy temperature is not reflected in Table2.Line 331,“wasobserved” should read “was observed”.

R27: Done. Modification was reported in the text. Please see L294 and L295

C28: Line 311,“wasobserved“should read ”was observed“.

R28: Done. Modification was reported in the text. Please see L315

C29: Line 331, treatment under N-100 also does not conform to general rules.

R29: Done. We agree with you. This was added in the text while it was become moderate and high N application. Modification was reported in the text. Please see L336

C30: Line 344: is the "+" in "+5,65, +9.91" necessary?

R30: Done. The "+" was removed from the text. Modification was reported in the text. Please see L352

C31: In Table 1 and 2,“values are the mean ± error standard” does not appear,please add.

R31: Done. The sentence was added in the title of table 1 and 2. Modification was reported in the text. Please see L175-177 and L304-307.

C32: In Table 3 and 5, the arrangement of the data is different from that in Table 2 and 4.

R32: Done. The data arrangement was homogenized among all tables. Modification was reported in table 1, 2, 3, 4 and 5. Please see cited tables.

C33: Line 346-347, incorrect use of punctuation.

R33: Done. Punctuations were corrected in title of table 4. Modification was reported in table 1, 2, 3, 4 and 5. Please see cited tables.

C34: In Table1 and Table2, there is no description values are the mean ± error standard.

R34: Done. The sentence was added for table 1 and 2. Modification was reported in table 1 and 2. Please see table 2.

C35: Table 2,“Wheat intercrop ”and “Chickpea intercrop ”What exactly does that mean? Aren’t they in conflict? Because there are only three planting factors in 2.2.

R35: In this case of results, the two parameters are measured in both intercropped species separately because no possibility to measure crop temperature simultaneously for both intercropped species, we measure these two parameters for each intercropped species. So wheat intercrop mean wheat intercropped with chickpea.

C36: Table 4,“0,01”The comma here should be replaced by the decimal point.

R36: Done. The comma was replaced by point. Modification was reported in table 4. Please see table 4.

C37: Table 5,line 367,“water use efficiency for whole biomass (WUEbh) and water use efficiency for grain yield (WUEgy) in chickpea”should be changed “water use efficiency for whole biomass (WUEYB) and water use efficiency for grain yield (WUEGY) in chickpea”

R37: Done. The two words were changed as your recommendation. Modification was reported in the title of table 5. Please see L375-376

C38: Table 5,“3107± 33.8bc”should be modified“3107± 33.8bc

R38: Done. Modification was reported in the table 5. Please see table 5.

Discussion

C39: Line 388,“(Figs. 3 and 4)”should be modified “(Table.3 and 4)”.

R39: Done. Figures were replaced by tables. Modification was reported in the text.  Please see L397

C40: Line 390,“sem-iarid (S3 site) climate”should be changed “semi-arid (S3 site) climate ”.

R40: Done. Modification was reported in the text.  Please see L399

C41: Lines 400–402: is there any basis for the inference that "Such improvements that are probably due to a higher ability of intercropped chickpea to increase N2-fixation was also associated with a significant increase of chlorophyll content (Table 2) only in leaf of intercropped"?

R41: Done. Thanks for this comment, sure we agree with you. As that, this sentence was reformulated in “Such improvements that are probably due to high N2-fixation by intercropped chickpea was also associated with a significant increase of chlorophyll content (Table 2). This was particularly demonstrated in leaf of intercropped durum wheat under low N-application. Modification was reported in the text.  Please see L409-412

C42: Line 443, there is a space before the new sentence.

R42. Yes offcourse, the space exist before the new sentence

Conclusion

C43: 44. Line 458: the conclusion was written comprehensively, but there was too much content and the main conclusions were not highlighted enough.

R43. Done. The main conclusion was rephrased to beter highlighted our major finding in this study. Modification was reported in the text.  Please see L512-515

References

C44: There are numerous problems with the format of the references, such as wrong symbols in the authors, italicized issue numbers and journal names, etc.

R44. Done. All references were checked according to agriculture journal recommendation. Please see the reference list.

C45: Line 525, there are several errors in the references, for example, the year of the research is not bold, etc.

R44. Done. This reference was corrected as all references. Please see the reference list in L552

Round 2

Reviewer 1 Report

Dear Authors,

thank you very much for addressing the comments I sent you and completing the paper.

In my opinion, the changes made have improved the quality of the manuscript, and clarified many issues.

However, I believe that a few more additions should be made before publishing the work.

Line 175 Thank you for explaining why the content of P and K is not given. However, I believe that they should be given. Deficiency or high availability of one mineral can affect the plant's uptake of other minerals. Even if the contents of P and K did not vary much between locations their values in Table 1 should be given. If tests for other minerals were also done please give them as well. This will certainly not divert attention from the main issue, which is N availability, and will allow other researchers and farmers to relate the experiment conducted by the authors to their own conditions.

Line 198 Thank you for the clarification. However, I often use pre-emergence preparations when growing legume-cereal mixtures. However, I agree that there could be interactions with N fertilizer. Please add to the manuscript, at this point, information about manual weed control.

Line 194 - 197 Thank you for the added information on the timing of N applications. Please also state in what amount N fertilizer was applied in these two treatments. (How much or what proportion of N was applied in the first treatment and how much in the second treatment)

Line 197. I repeat the question from the first review about the application of other fertilizers (P and K). If they were applied necessary please include this in the manuscript for reasons analogous to the soil mineral content.

Line 474 - 478 Very good and accurate statement.

Line 414 - 422. also a very important point of discussion and very aptly described.

Please work more on the reference section. The DOI address is missing in several places (even though it is available) among others number 8, 15, 16 and so on.

Reference 16 is missing one of the authors.

Still in several places journal names are given by abbreviation and some by full name. Please standardize (for example, 24, 27)

Number 36 should read: Pużyńska, K.; Pużyński, S.; Synowiec, A.; Bocianowski, J.; Lepiarczyk, A. Grain Yield and Total Protein Content of Organically Grown Oats-Vetch Mixtures Depending on Soil Type and Oats' Cultivar. Agriculture 2021, 11, 79. https://doi.org/10.3390/agriculture11010079

Please consider the positives in the manuscript in the section discussing the impact of high or low NUE on post-harvest soil of the main crop and the possible impact on succeeding crops. This is not the main issue addressed in the paper but I think it would provide some supplementation and suggestions for other researchers and farmers. (This is just a suggestion)

Author Response

Reviewer 1

Comments and Suggestions for Authors

Dear Authors,

Thank you very much for addressing the comments I sent you and completing the paper. In my opinion, the changes made have improved the quality of the manuscript, and clarified many issues.However, I believe that a few more additions should be made before publishing the work.

Thanks very much for the comment. This improvement was occur thanks to your valuable comments.

C1. Line 175: Thank you for explaining why the content of P and K is not given. However, I believe that they should be given. Deficiency or high availability of one mineral can affect the plant's uptake of other minerals. Even if the contents of P and K did not vary much between locations their values in Table 1 should be given. If tests for other minerals were also done please give them as well. This will certainly not divert attention from the main issue, which is N availability, and will allow other researchers and farmers to relate the experiment conducted by the authors to their own conditions.

R1: Done. All values of N, P and K available were added in table 1. Indeed we give all interpretation of these three added results (N, P and K) in the text. Please see table 1 and L174-178

C2. Line 198 Thank you for the clarification. However, I often use pre-emergence preparations when growing legume-cereal mixtures. However, I agree that there could be interactions with N fertilizer. Please add to the manuscript, at this point, information about manual weed control.

R2: Done. Manual weeding was added in the text. Please see L204

C3. Line 194 - 197 Thank you for the added information on the timing of N applications. Please also state in what amount N fertilizer was applied in these two treatments. (How much or what proportion of N was applied in the first treatment and how much in the second treatment)

R3: Done. The quantities that were applied in relation with time of N-application were clarified in the text. Please see L200-202

C.4. Line 197. I repeat the question from the first review about the application of other fertilizers (P and K). If they were applied necessary please include this in the manuscript for reasons analogous to the soil mineral content.

R4. Dear reviewer, in our experiment we applied only N fertilizer treatment, no application of P and K fertilizers.

C5. Line 474 - 478 Very good and accurate statement.

R5: Thanks very much.

C.6. Line 414 - 422. also a very important point of discussion and very aptly described.

R6: Thanks very much.

C.7. Please work more on the reference section. The DOI address is missing in several places (even though it is available) among others number 8, 15, 16 and so on.

R7: Done. All missing DOI are now completed in all references. Please see reference list.

C.8. Reference 16 is missing one of the authors.

R8: Done. Thanks, Pease see reference 16 that is now corrected by adding the missing author.

C.9. Still in several places journal names are given by abbreviation and some by full name. Please standardize (for example, 24, 27)

R9: Done. All journal names are standardized within full name journal for all cited references. Please see the list of reference.

C.10. Number 36 should read: PużyÅ„ska, K.; PużyÅ„ski, S.; Synowiec, A.; Bocianowski, J.; Lepiarczyk, A. Grain Yield and Total Protein Content of Organically Grown Oats-Vetch Mixtures Depending on Soil Type and Oats' Cultivar. Agriculture 2021, 11, 79. https://doi.org/10.3390/agriculture11010079

R10: Done. Thanks very much. This reference was replaced by that correcte. Please see the new reference 36.

C.11. Please consider the positives in the manuscript in the section discussing the impact of high or low NUE on post-harvest soil of the main crop and the possible impact on succeeding crops. This is not the main issue addressed in the paper but I think it would provide some supplementation and suggestions for other researchers and farmers. (This is just a suggestion).

R11 : I am 100% with you dera reviewer and if you see our conclusion you will find that we propose this intercropping system as rotated system with durum wheat. We propose to replace legume monoculture in rotation durum wheat-legumes by rotation durum intercropping/durum wheat/intercopping because intercropping is more stable in terms of yield and resources use efficiency.

Reviewer 2 Report

Thanks for revising the paper. I have checked the revision and I am satisfied with the changes you made to this manuscript.

Author Response

Thanks dear reviewer for your aknowgment